# Viewpoint-Agnostic Grasp Pipeline using VLM and Partial Observations

*Abstract*— **Robust grasping in cluttered, unstructured environments remains challenging for mobile legged manipulators due to occlusions, unreliable depth, and the need for collision-free, execution-feasible approaches. We present an end-to-end pipeline for language-guided grasping that bridges open-vocabulary target selection to safe grasp execution on a real robot. Given a natural-language command, the system grounds the target in RGB using open-vocabulary detection and promptable segmentation, extracts an object-centric point cloud from RGB-D, and improves geometric reliability under occlusion via back-projected depth compensation and two-stage point cloud completion. We then generate and collision-filter 6-DoF grasp candidates and select an executable grasp using safety-oriented heuristics that account for reachability, approach feasibility, and clearance. We evaluate the method on a Boston Dynamics Spot with an arm in two cluttered tabletop scenarios, using paired trials against a view-dependent baseline. The proposed approach achieves a 90% success rate (9/10) versus 30% (3/10) for the baseline, demonstrating substantially improved robustness to occlusions and partial observations in clutter.**

## I. INTRODUCTION

Robust object grasping in cluttered environments remains a fundamental challenge for autonomous robotic manipulation. In real-world deployments such as inspection, remote intervention, and field operations, robots must interact with partially observed objects under severe occlusions, limited viewpoints, and incomplete depth measurements [1], [2], [3]. Successful manipulation in these conditions requires more than a geometrically valid grasp: the selected grasp must admit a collision-free approach, respect kinematic constraints, and remain stable during execution [4], [5], [6]. Grasps that appear feasible on visible surfaces often become unreliable once hidden geometry, approach trajectories, and physical interaction constraints are considered [7], [8].

Mobile manipulation platforms, including quadruped robots equipped with manipulators, provide the mobility and perception needed for such environments [9], [10], [11], [12]. Most recent grasping pipelines predict 6-DoF grasps directly from depth or point clouds [13], [14], [15]: GPD samples and ranks 6-DoF candidates from partial observations [13], [16]; Contact-GraspNet improves grasp quality in clutter by treating grasping as contact-point classification [14]; and AnyGrasp leverages dense point-based predictions across multiple views and time steps [15]. Complementary 3D encodings such as feature splatting [17] and Gaussian-splatting-based real2sim2real transfer [18] have also been explored. While these methods improve grasp perception, mobile manipulation still requires converting partial, instance-level observations into executable grasps with feasible and secure approach trajectories.

In open-world deployments, the target object is often specified semantically rather than pre-segmented. Vision-language models (VLMs) and language-conditioned perception provide scalable interfaces for task-driven target selection in cluttered scenes [24], [25]. Promptable open-set detectors such as Grounding DINO [19], combined with segmentation models such as SAM 2 [20], enable text-conditioned detection and instance segmentation from RGB observations [26]. However, bridging semantic grounding to reliable 3D grasp execution under partial observation remains challenging: the robot must convert grounded masks into object-centric geometry, infer missing surfaces, and generate grasps that remain feasible under real motion and safety constraints. Existing works typically address perception, grasp prediction, or execution individually rather than as a unified execution-aware pipeline.

In this paper, we address end-to-end robust grasping in clutter for mobile legged manipulation, from natural-language-driven target selection to safe execution on a real robot under partial observations. We integrate open-vocabulary target selection with object-centric 3D estimation and spatial-aware grasping in a unified pipeline, as shown in Fig. 1. Target selection is grounded via VLM-guided detection and segmentation; geometry is derived from RGB-D with back-projected depth compensation and two-stage point cloud completion; and grasps are selected for safe execution using heuristics that account for approach feasibility, clearance, and reachability. The main contributions are: **(i)** a unified pipeline bridging natural-language target specification and execution-feasible grasping for mobile legged robots; **(ii)** an occlusion-resilient geometry estimation module combining depth back-projection with MGPC and PoinTr completion; **(iii)** an execution-aware grasp selection strategy incorporating approach, clearance, and reach constraints, coordinated with base repositioning; and **(iv)** real-world validation on a Boston Dynamics Spot in cluttered tabletop scenes, showing a 90% success rate versus 30% for a view-dependent baseline.

## II. METHODS

The proposed pipeline consists of four modules: (A) detection and segmentation, (B) point cloud generation and estimation, (C) grasp pose generation and selection, and (D) execution. The framework (Fig. 1) takes RGB-D images from the robot's cameras as input and performs grasping and locomanipulation in cluttered environments. It is implemented in ROS 2 [27], and the code will be made available in our repository.

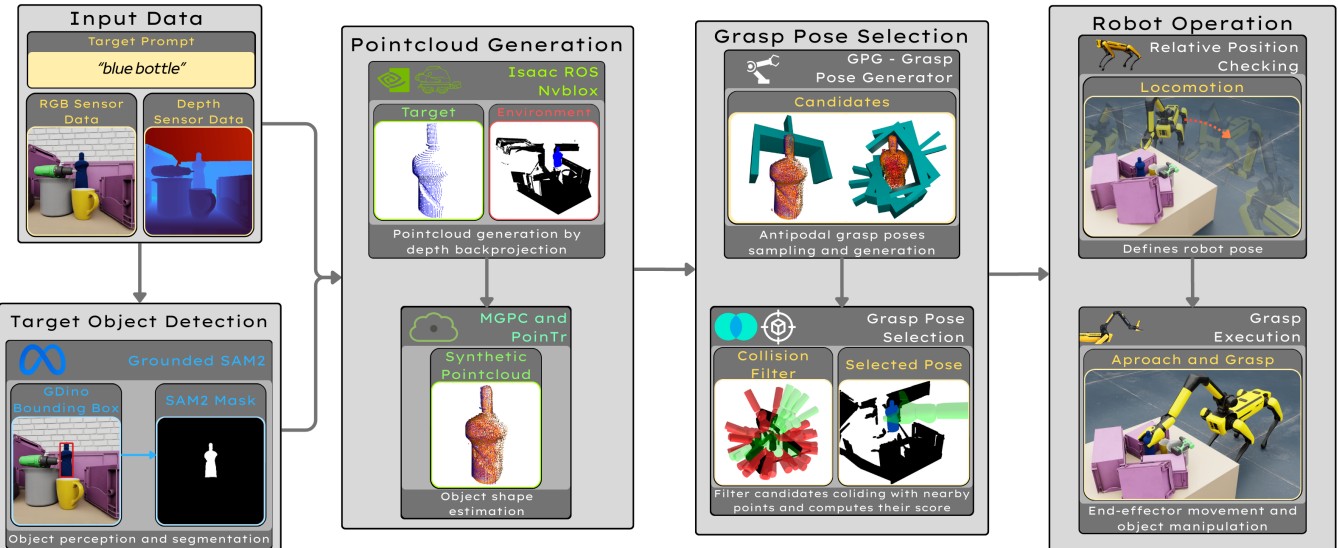

Fig. 1: System overview of the proposed viewpoint-agnostic grasping pipeline. The system receives a natural-language target prompt (e.g., "blue bottle") together with synchronized RGB and depth observations. The prompt is grounded in RGB using Grounding DINO [19] to obtain a target bounding box and SAM 2 [20] to produce an instance mask. The mask is then used to extract an object-centric partial point cloud from depth via Isaac ROS Nvblox [21] with depth backprojection. To mitigate occlusions and sparse depth, the object geometry is completed in two stages: MGPC [22] generates synthetic points conditioned on the prompt, RGB, and the partial point cloud, and PoinTr [23] further densifies the geometry by completing fixed-size local patches. Given the densified point cloud, GPG [16] samples antipodal 6-DoF grasp candidates, which are collision-filtered and ranked to select an execution-feasible grasp. Finally, the robot executes a state-machine locomanipulation routine that repositions the base for reachability and clearance, followed by end-effector approach, grasp closure, and object lift.

## A. Detection and Segmentation

The perception stage takes RGB images from the robot's front cameras as input and outputs an object-centric instance mask used to extract 3D geometry (Sec. II-B). The robot provides stereo RGB-D data at 15 Hz; detection and segmentation use RGB only, while depth is used in Sec. II-B via nvblox-based depth integration [21].

The operator specifies the target via a natural-language command (e.g., "blue bottle"). The target is localized using Grounding DINO [19], which returns candidate bounding boxes with confidence scores; we select the highest-scoring box $B^\star$ and use it to initialize SAM 2 [20] for instance segmentation and video tracking. During execution, SAM 2 maintains the target mask across frames, and Grounding DINO is re-invoked only if tracking fails. We pass $B^\star$ as a box prompt to SAM 2 to obtain a binary mask $M$, which is refined via a lightweight morphological erosion (OpenCV) to suppress boundary leakage into nearby clutter. If no valid hypothesis is produced, the system does not proceed to grasp planning and continues acquiring observations.

## B. Point Cloud Generation and Estimation

This stage converts the mask $M$ into object-centric 3D geometry suitable for grasp synthesis under partial observations. We leverage Isaac ROS nvblox for GPU-accelerated depth processing and point cloud extraction [21]. In contrast to volumetric TSDF/ESDF mapping, our pipeline operates

directly on point clouds with back-projected depth compensation to reduce sparsity and missing returns.

For each RGB-D frame, nvblox back-projects the depth image and produces a registered scene point cloud in the robot frame. Applying $M$ retains only points that project inside the segmented target, yielding an object-centric partial point cloud $P_{\text{partial}}$. Masked clouds are aggregated over time using the robot's state estimation, increasing surface coverage while remaining lightweight. Depth in clutter often contains holes, flying pixels near discontinuities, and missing returns on thin or specular structures; we mitigate these by filling small holes and attenuating outliers using local neighborhood consistency in the image plane before extracting $P_{\text{partial}}$.

Even after multi-frame accumulation and depth compensation, $P_{\text{partial}}$ remains incomplete due to self-occlusions and backside surfaces. We apply MGPC [22] to estimate missing geometry by leveraging multimodal context (prompt, RGB, partial cloud). MGPC requires a fixed-size input, so we subsample $P_{\text{partial}}$ to 2048 points and infer a synthetic set $P_{\text{mgpc}}$ of 8192 points. Observed and synthetic points are merged as

$$P_{\text{mid}} = P_{\text{partial}} \cup P_{\text{mgpc}}, \tag{1}$$

increasing surface coverage while remaining consistent with visible geometry. Since the Grasp Pose Generator (GPG) is sensitive to normal estimation quality, we further densify $P_{\text{mid}}$ using PoinTr [23]. We decompose $P_{\text{mid}}$ into overlapping

2048-point patches via a KD-tree neighborhood query; each patch is completed independently, and the union of completed patches is merged with $P_{\text{mid}}$ to yield the final cloud $P_{\text{complete}}$ (typically from $\sim$2k to $\sim$10k points after MGPC, and more after PoinTr). $P_{\text{complete}}$ is then passed to the grasp module.

*C. Grasp Pose Generation and Selection*

Given $P_{\text{complete}}$, we sample 1000 candidate 6-DoF grasps using GPG [16] and select a single grasp $g^\star$. Densifying $P_{\text{complete}}$ improves normal stability and increases the diversity of feasible hypotheses. Each candidate $g_i = (\mathbf{p}_i, \mathbf{R}_i)$ defines a gripper pose with position $\mathbf{p}_i$ and orientation $\mathbf{R}_i$ in the robot frame. GPG's antipodal sampling parameters (jaw width and contact constraints) are configured to match the Spot jaw gripper geometry.

To enforce feasibility in clutter, each candidate is validated by collision checking: we test the gripper mesh at pose $g_i$ against the environment point cloud in a neighborhood around the target using a parallelized kernel. Colliding candidates are rejected, yielding a filtered set $\mathcal{G}_{\text{free}} \subseteq \mathcal{G}$. From $\mathcal{G}_{\text{free}}$, we select the grasp that best trades off approach feasibility and grasp stability via the weighted cost

$$\mathcal{C}(g_i) = w_\theta |\Delta \theta_i| + w_\phi \phi_i + w_c \|\mathbf{p}_i - \mathbf{c}\| + \mathcal{P}(r_i), \quad (2)$$

where $\mathbf{c}$ is the centroid of $P_{\text{complete}}$ and $r_i = \|\mathbf{p}_i - \mathbf{p}_{\text{base}}\|$. The alignment term $|\Delta \theta_i|$ is the angular deviation between the base-to-target direction and the grasp approach direction (biasing toward the nominal approach and less repositioning); $\phi_i$ is a binary penalty discouraging unfavorable approaches (e.g., from below), which are more likely to be kinematically constrained or blocked in clutter; $\|\mathbf{p}_i - \mathbf{c}\|$ favors centered grasps that are less sensitive to partial geometry; and $\mathcal{P}(r_i)$ is a hard reach penalty,

$$\mathcal{P}(r_i) = \begin{cases} 0, & r_i \leq r_{\max}, \\ M, & r_i > r_{\max}, \end{cases} \quad (3)$$

with $M$ a large constant. The final grasp is $g^\star = \arg\min_{g_i \in \mathcal{G}_{\text{free}}} \mathcal{C}(g_i)$. In hazardous environments, the weights $(w_\theta, w_\phi, w_c)$ and $r_{\max}$ provide explicit parameters to tune grasp selection toward safer, more reliable grasps.

*D. Grasp Execution and Motion Control*

Execution of $g^\star$ is managed by a finite-state machine that coordinates base repositioning and arm motion to ensure reachability. If $g^\star$ is not reachable from the current stance, the robot commands a base waypoint along the grasp approach direction at a stand-off distance that improves manipulability while keeping the target within the arm workspace, satisfying reachability and clearance constraints before arm motion. After base alignment, a pre-grasp pose $g_{\text{pre}} = g^\star \oplus (\delta\,\hat{\mathbf{x}})$ is computed by offsetting $g^\star$ along the gripper approach axis $\hat{\mathbf{x}}$ by a safety distance $\delta$, where $\oplus$ denotes pose composition. From $g_{\text{pre}}$, the end-effector executes a short Cartesian insertion of length $\ell = 5\,\text{cm}$ along the approach axis to reach $g^\star$; the gripper service then closes the jaws to secure the object.

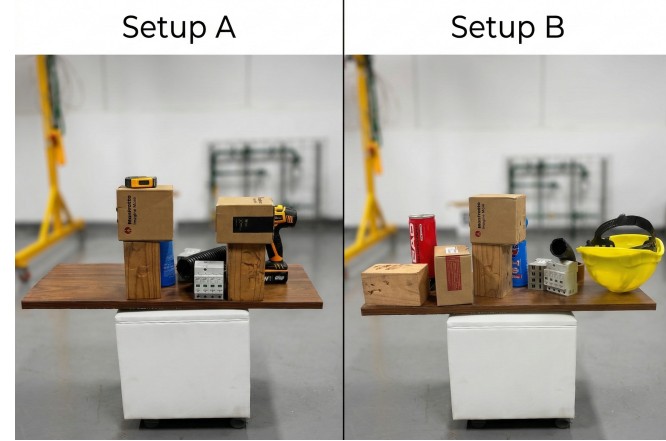

Fig. 2: Experimental setups. Setup A (left): grasp a power drill partially obscured by boxes and electrical components. Setup B (right): grasp a blue bottle situated behind different boxes. Both configurations test the pipeline's ability to generate feasible grasps under occlusion and clutter.

## III. EXPERIMENTS

We evaluate the proposed pipeline on a Boston Dynamics Spot equipped with a 6-DoF arm and jaw gripper, in cluttered scenes representative of unstructured deployments. The goal is to quantify the benefits of geometry estimation under partial observations and spatial-aware grasp selection, compared with a view-dependent baseline that fixes the robot's position upon detection.

*A. Hardware, Software, and Protocol*

Experiments are run on Spot with its onboard RGB-D sensing; computation is split between the robot and an external workstation connected via Ethernet. Spot's front-left and front-right cameras provide RGB at VGA resolution ($640 \times 480$, 15 Hz) and stereo depth at $424 \times 240$. The workstation ($2 \times$NVIDIA RTX A2000 12 GB, Intel® Xeon® w3-2435, 125 GB RAM) runs the learning-based components (Grounding DINO, SAM 2, MGPC, PoinTr) and point-cloud processing. Robot communication uses the Spot SDK, and the full system runs end-to-end in real time under ROS 2 (Docker, Humble/Jazzy). The finite-state machine in Sec. II-D drives execution.

*B. Setups, Baseline, and Protocol*

Two cluttered tabletop setups are evaluated (Fig. 2), differing mainly in the target: Setup A (handheld drill) and Setup B (blue bottle), each surrounded by distractors and occluders. The table height is comparable to the robot base, allowing consistent visibility from the front cameras, and the robot faces the table at the start of each trial. We compare our full pipeline (mask-conditioned extraction, multi-frame accumulation, MGPC+PoinTr completion, and mobile grasp selection/execution) against a **view-dependent baseline** that shares the same perception front-end and GPG candidate generation, with identical collision filtering and heuristic ranking, but plans directly on the single-view partial cloud

| | | | Success or Failure (alongside failure reason) | |
|---|---|---|---|---|
| Scenario | Run | Object | Our Method | Baseline (View-Dependent) |
| A | 1 | drill | ✓ | ✗ (reachability failure) |
| | 2 | drill | ✓ | ✗ (approach collision (clutter)) |
| | 3 | drill | ✓ | ✗ (approach collision (clutter)) |
| | 4 | drill | ✗ (reachability failure) | ✗ (reachability failure) |
| | 5 | drill | ✓ | ✗ (approach collision (clutter)) |
| B | 1 | blue bottle | ✓ | ✗ (approach collision (target)) |
| | 2 | blue bottle | ✓ | ✓ |
| | 3 | blue bottle | ✓ | ✗ (approach collision (clutter)) |
| | 4 | blue bottle | ✓ | ✓ |
| | 5 | blue bottle | ✓ | ✓ |
| Total success rate | | | (9/10) | (3/10) |

TABLE I: Success/failure outcomes across scenarios A and B. The failure reason is presented whenever the robot fails to either reach or securely grasp the target object.

without multi-frame accumulation or completion, and executes from the initial stance without base repositioning. This baseline isolates the impact of viewpoint-agnostic geometry estimation and mobile execution, reflecting a common deployment setting in existing grasp pipelines.

Each setup comprises 10 trials (5 per method) using a paired protocol: for each initial robot pose, one trial is run with our method and one with the baseline from the same position. Across pairs, the initial robot position is varied (with the robot still facing the setup) to test robustness to viewpoint-dependent occlusion and depth sparsity. After each attempt, the robot returns to its initial pose. A trial is labeled **successful** if the robot (i) closes the gripper on the target, (ii) lifts it from the table, and (iii) maintains a stable grasp for a short verification period without dropping or slipping. For reproducibility, perception thresholds, completion settings, and grasp-selection weights are documented in the code repository.

## IV. RESULTS AND DISCUSSION

Table I summarizes outcomes across the two scenarios. Our method achieves **9/10** successful trials, versus **3/10** for the view-dependent baseline. In Scenario A (drill), our pipeline succeeds in **4/5** trials while the baseline fails in all **5/5**; in Scenario B (blue bottle), our method succeeds in **5/5** and the baseline in **3/5**. Incorporating partial-observation geometry estimation and completion substantially improves robustness to clutter and occlusions in both setups. A video of the experiments can be found at the following link:

Conducted experiments

### A. Failure Mode Analysis

Failures fall into three categories: **FM-1 (reachability)**, where no feasible pre-grasp configuration exists within the arm workspace; **FM-2 (approach collision, target)**, where collision occurs while approaching the target due to limited clearance; and **FM-3 (approach collision, clutter)**, where collision occurs with surrounding occluders during approach. The baseline fails predominantly due to approach collisions (FM-2/FM-3) in both scenarios, indicating that the initial view-dependent partial cloud produces grasp candidates that

are locally plausible but not executable once approach clearance is considered. Our method leaves only a single failure across all trials (Scenario A, run 4), categorized as FM-1. This supports the hypothesis that improved object-centric geometry under partial observations (depth compensation + completion) increases the number of grasps remaining feasible under collision and reachability constraints.

### B. Key Observations and Limitations

Three observations emerge: (1) object-centric geometry estimation with partial-observation completion increases grasp candidate reliability under occlusion; (2) execution-aware grasp selection with collision-aware filtering keeps grasps feasible during real execution, reducing approach collisions common when planning relies solely on view-dependent geometry; and (3) integrating mobile base repositioning with grasp planning improves accessibility by satisfying reachability and approach constraints before arm execution. Limitations remain: the target must be sufficiently within the robot's field of view for open-vocabulary grounding to reliably identify the intended instance, ambiguous prompts or heavy occlusion can cause mis-grounding, domain-specific objects may require task-specific prompt engineering or fine-tuning, and severe depth noise or limited sensor resolution can still degrade geometry estimation for thin or reflective objects, where completion may not sufficiently recover the correct shape.

## V. CONCLUSION

We presented an end-to-end pipeline for language-guided, viewpoint-agnostic grasping in clutter on a legged mobile manipulator. The system grounds a natural-language target via open-vocabulary detection and promptable segmentation, builds object-centric geometry from RGB-D with depth compensation and two-stage point cloud completion, and selects 6-DoF grasps using execution-aware heuristics executed by a state-machine controller on Spot. Experiments in two cluttered tabletop scenarios show substantial reliability gains over a view-dependent baseline, supporting the hypothesis that robust grasping benefits from bridging semantic grounding, object-centric 3D estimation, and execution-feasible selection.

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
