# OpenReview forum: "Viewpoint-Agnostic Grasp Pipeline using VLM and Partial Observations"
_IEEE.org/ICRA/2026/Workshop/Manipulation_Robustness — ICRA 2026_

### Official Review · Reviewer_V9jj · 2026-05-03

**Rating:** 6
**Confidence:** 4

**Review:**

This paper proposes a view-agnostic grasping pipeline for mobile manipulation in cluttered scenes. The system grounds a target object from a natural-language command, reconstructs its partial 3D geometry, and selects an execution-feasible grasp under collision and reachability constraints. The authors evaluate the system in real-world experiments on two scenes against an ablated view-dependent baseline.

Some of my concerns:

1. Several pipeline details would benefit from further clarification:
  - In Section II-B, how are small holes filled and outliers attenuated in the image plane?
  - How is SAM 2 tracking failure detected, and how does the system recover to maintain target tracking?
  - How are camera-to-robot poses obtained, calibrated, and synchronized for multi-frame point-cloud aggregation?
  - If the selected grasp pose is not reachable, how is the base waypoint generated, and how is the repositioning motion executed?
2. The paper lacks comparison with closely related grasping methods [1] [2]. Including these relevant baselines would significantly strengthen the empirical evaluation.
3. The grasp selection strategy relies heavily on hand-tuned heuristics, which may limit generalization to more diverse scenes.

References

[1] Lemke et al. *Spot-Compose: A Framework for Open-Vocabulary Object Retrieval and Drawer Manipulation in Point Clouds*, ICRA Workshop 2024.

[2] Kashyap et al. *Single-View Shape Completion for Robotic Grasping in Clutter*, RiTA 2025.

---

### Decision · Program_Chairs · 2026-05-21

Accept